# The Perceived Impact of COVID-19 on Comfort Food Consumption over Time: The Mediational Role of Emotional Distress

**DOI:** 10.3390/nu13061910

**Published:** 2021-06-02

**Authors:** Camila Salazar-Fernández, Daniela Palet, Paola A. Haeger, Francisca Román Mella

**Affiliations:** 1Departamento de Psicología, Universidad de La Frontera, Temuco 478000, Chile; camila.salazar@ufrontera.cl (C.S.-F.); daniela.palet@ufrontera.cl (D.P.); 2Departamento de Ciencias Biomédicas, Facultad de Medicina, Universidad Católica del Norte, Coquimbo 1781421, Chile

**Keywords:** COVID-19 pandemic, psychological distress, emotional eating, structural equation modeling, longitudinal

## Abstract

The COVID-19 pandemic has had a significant impact on populations at an economic, health, and on an interpersonal level, it is still unclear how it has affected health-risk behaviors, such as comfort food consumption over time. This study longitudinally examines the effect of the perceived impact of COVID-19 on comfort food consumption and whether this effect is mediated by emotional distress. A convenience sample of 1048 students and university staff (academic and non-academic) from two universities completed monthly online surveys during the COVID-19 pandemic across six waves (W; W1 to W6). Participants reported their perceived impact of COVID-19 (economic, interpersonal, and health), comfort food consumption, and emotional distress (DASS-21). Using structural equation models, we found an indirect longitudinal effect of the perceived impact of COVID-19 (W1) on comfort food consumption (W3 to W6) through increased emotional distress (W2). The perceived negative impact of COVID-19 on comfort food consumption was fully mediated by the emotional distress during the first waves (W3 and W4), ending in a partial mediation in the last waves (W5 and W6). These findings contribute to disentangling the mechanisms by which the perceived impact of COVID-19 affects comfort food consumption over time, and highlight the role of emotional distress. Future interventions should address comfort food consumption by focusing on handling emotional distress during a crisis.

## 1. Introduction

The COVID-19 pandemic declared in March 2020 has impacted public health, due to its high contagion and death rates. To reduce its transmission, social distancing measures have been implemented, affecting various aspects of daily life [1,2,3,4]. These consequences have been reported at several levels. At the economic level, most countries have shown higher unemployment rates, job insecurity, and lower wages [5]. At the interpersonal level, social interactions between family, work, and friends have been restricted due to lockdowns, and as a consequence, social support has been reduced [6,7]. At the health level, detrimental physical effects have been experienced by people with the SARS-CoV-2 virus, but lockdowns have also led to an increase in mental health problems [8,9]. Due to the multiple effects of COVID-19 on everyday life, several studies have reported an increase in adverse emotional responses, such as stress, anxiety, fear, and worry [1,10].

The constant perception of threat and experiencing emotional distress during the COVID-19 pandemic have been associated with adopting coping behaviors that allow people to respond to stress and uncertainty, which are not always associated with healthy behaviors [1,3,11]. Thus, several studies have reported that during the COVID-19 pandemic, people have adopted unhealthy behaviors to deal with high levels of stress, such as alcohol [12,13], cigarettes [14], and food consumption [15,16]. The latter has been one of the most critical aspects, since changes in people’s eating patterns usually occur in the short- and long-term during crises, generating malnutrition or overnutrition problems [17].

The prolonged time of confinement and the long working routines during the COVID-19 pandemic has impacted daily eating practices, showing a trend towards greater consumption of processed foods, such as convenience meals, junk food, and snacks, especially desserts, chocolate, ice creams, and salty snacks [15,18,19,20,21]. At a biochemical level, consuming foods high in carbohydrates, salts, fats, and sugars has shown associations with a stress-reduction strategy by increasing serotonin production, which positively affects mood [22,23]. Using comfort food as a response to negative emotions (i.e., emotional eating) has been identified by the literature as a coping mechanism in the face of acute stressors [16,24,25,26].

It is expected that consuming comfort food might be a mechanism to deal with the psychological distress produced by the COVID-19 pandemic [27,28]. Specifically, research has shown that to counteract the negative experience of isolation and boredom that can arise from staying at home for an extended period; people are likely to seek an escape from monotony by consuming more comfort food [18,29,30]. This continues even when there are signs of satiety [31]. Adopting this coping strategy has negative health consequences, since it increases the risk of developing obesity and cardiovascular disease and generates a chronic level of inflammation. Such inflammation is associated with a greater risk of severe complications due to COVID-19 [32,33].

This study will examine longitudinally the relationship between the perceived economic, interpersonal, and health impact of COVID-19 on food consumption over time and whether this association is mediated by emotional distress. The mediation hypothesis is supported by the Lazarus and Folkman stress and coping model [34]. This model suggests that recognizing a threat (i.e., the COVID-19 pandemic) initiates a process that activates negative emotional reactions that may result in unhealthy coping mechanisms (e.g., consumption of comfort food). The mediation effect has been found at a cross-sectional level in the context of infectious diseases similar to SARS-CoV-2, such as H1N1. The findings have suggested that threats of being infected trigger both the experience of negative emotions (i.e., anxiety, worry, fear) and engagement in coping behaviors [35,36,37]. Moreover, Ramalho et al. [38] explored cross-sectionally the impact of COVID-19 lockdowns on disordered eating behaviors through the mediating effect of psychological distress. To our knowledge, no previous study has longitudinally examined the potential mediating effect of emotional distress in the relationship between the perceived impact of the pandemic and comfort food consumption as reactions to such stress. Furthermore, such mediation has not been evaluated longitudinally to explore if it is stable over time. In particular, we hypothesized that the threat-emotion-coping sequence proposed by Lazarus and Folkman might be more robust at the beginning of the pandemic. At this time, the threat and impact of COVID-19 on people’s lives were greater, leading people to experience more emotional distress, and therefore, develop coping mechanisms [39]; in this case, eating more comfort food. Accordingly, we expect that this mediation effect decreased over time, due to chronic exposure to the pandemic and the development of emotion regulation processes [34].

## 2. Materials and Methods

### 2.1. Participants and Procedure

Participants took part in a panel study that aimed to assess the impact of COVID-19 on university staff (academic and non-academic) and students’ mental health from two Chilean universities. Data were collected during July (W1, *n* = 1038), August (W2, *n* = 509), September (W3, *n* = 412), October (W4, *n* = 475), November (W5, *n* = 430) and December (W6, *n* = 415) 2020. The participants in this study were aged between 18 and 73 years (*M* = 29.52, *SD* = 11.66), 69% were female, and 66.3% were university students. Participants were informed of the nature of this study and signed an informed consent indicating their willingness to participate with no incentive for participation in any of the study waves. Although the number of participants decreased in the subsequent waves after W1, the age average and the proportion of women and students remain stable (see Appendix A). In both universities, the Scientific Ethics Committee approved this study (resolutions 086/20 on July 2020 and 11/2020 on June 2020, respectively).

### 2.2. Measures

Data for this study were collected using online surveys due to the current COVID-19 pandemic. Completing the entire questionnaire took approximately 10–15 min. Only the variables analyzed in the present study are detailed.

#### 2.2.1. Predictor: COVID-19 Perceived Impact

Three-items evaluated the perceived negative impact of COVID-19 at the economic level, interpersonal relationships with family and friends, and own and friends’ health in W1. A 5-point response scale was used (0 = not at all, 1 = a little, 2 = some, 3 = quite a bit, 4 = a lot). High scores indicated a greater perceived negative impact of COVID-19. Model reliability showed appropriate construct consistency for a 3-items scale (Composite Reliability, (CR) = 0.597).

#### 2.2.2. Mediator: Emotional Distress (DASS-21)

We used the abbreviated 21-item scale by Lovibond and Lovibond [40], which assesses depression, anxiety, and stress separately (7 items each). We estimated the presence and intensity of negative emotional states experienced in the last week (i.e., emotional distress) in W2. For the supplementary analysis, we used measures of emotional distress from W3 to W6. The response format is a 4-point severity scale (0 = It did not happen to me, 1 = It happened to me at some point, 2 = It happened to me quite a lot, and 3 = It happened to me a lot). The depression, anxiety, and stress items were tallied to obtain the total score for each of these constructs. Composite reliability was an indicator of good construct consistency (CR = 0.884).

#### 2.2.3. Criterion: Past-Week Frequency of Comfort Food Consumption

A 5-item scale that evaluates the consumption of comfort food was used. This instrument asked participants across W3 to W6, the following: “During the last week, on how many days have you consumed: Fried meals (e.g., fried meat, fish, eggs, fries), sugary drinks (i.e., cola drinks, bottled juice), desserts or candies (e.g., ice cream, chocolate, candies, cakes, pastries), snacks (potato chips, chocolate bars, candy bar, cookies) and fast food (e.g., hamburgers, pizzas, hot dogs)”. These questions refer to foods that are considered comfort food, that is highly processed and rich in saturated fat, sugar, or sodium. The responses range from 0 to 7 days for each food. High scores indicate a higher frequency of consumption of comfort food. Model reliability showed acceptable construct consistency (CR = 0.719).

### 2.3. Statistical Analysis

First, descriptive analyses were carried out, and the distribution of the variables proposed in the hypothesized model was assessed. As expected, the data did not present a normal distribution at the uni-, bi-, and multi-varied levels, but we addressed this problem using a proper estimation method which we describe below. We estimated the reliability of the model for each of the measures using the CR. Cut-off values above 0.6 indicate appropriate construct consistency for instruments with more than three items [41,42]. Structural equation models (SEM) were used to assess whether the COVID-19 perceived impact had a longitudinal effect on comfort food consumption mediated by emotional distress. Data were analyzed using Stata 15.1 [43] using the full information maximum likelihood estimation method (FIML). This method uses all the available data to estimate the model parameters based on the information that the participants provided for the previous measures and is also robust against violations of the assumption of normality [44].

To respond to the primary objective of the study, we examined whether emotional distress is a mediator of COVID-19 perceived impact on consuming comfort food over time. According to Iacobucci [45], structural equation models are an excellent method to evaluate possible mediating effects with latent variables. To assess the mediation, we used the *medsem* package [46], which considers the estimation of mediating or indirect effects with bootstrapping techniques (samples = 5000) and the Monte Carlo test to estimate confidence intervals (CI) at 90% [47].

In the present study, four SEM models were estimated to evaluate the direct longitudinal impact of COVID-19 (W1) on comfort food consumption (W3 to W6) and mediated through emotional distress (W2). The SEM models were evaluated using the comparative fit index (CFI), the Tucker Lewis index (TLI), and the square root of the mean error of approximation (RMSEA) with CI at 90%. These indices were interpreted according to the conventional goodness of fit criteria: CFI and TLI > 0.95 and RMSEA ≤ 0.08 [48,49].

## 3. Results

First, the descriptive statistics and the correlations between COVID-19 perceived impact (in W1), emotional distress (in W2), and comfort food consumption (from W3 to W6) were analyzed and are presented in Table 1. Overall, the Pearson’s correlations coefficients between COVID-19 perceived impact and emotional distress with comfort food consumption over time were positive and ranged from small to moderate.

### 3.1. Measurement Models

Before evaluating the hypothesized SEM model, the fit of each of the constructs was tested through a confirmatory factor analysis (CFA). The first CFA was specified from two latent factors: (1) COVID-19 perceived impact (W1), which included perceived economic, interpersonal, and health impact of COVID-19, and (2) Emotional distress (W2), which included the items corresponding to the total scores of each scale (stress, anxiety, and depression). Because each of these factors (i.e., COVID-19 perceived impact and emotional distress) only has three items, statistical identification was not possible [50], and a model with two related factors was estimated. This model achieved an excellent fit, χ^2^ (8) = 4.848, *p* = 0.774, CFI = 1.000, TLI = 1.000, RMSEA = 0.000 [90% CI = 0.000, 0.025]. Finally, the evaluation of the measurement model for the comfort food consumption was estimated at W3, W4, W5, and W6. The estimate of the first CFA revealed that the comfort food consumption model presented excellent model fit across W3 to W6, W3, (χ^2^ (5) = 8.143, *p* = 0.149, CFI = 0.991, TLI = 0.982, RMSEA = 0.039 [90% CI = 0.000, 0.086]; in W4, χ^2^ (5) = 8.164, *p* = 0.147, CFI = 0.990, TLI = 0.979, RMSEA = 0.037 [90% CI = 0.000, 0.080]; in W5, χ^2^ (5) = 8.626, *p* = 0.125, CFI = 0.988, TLI = 0.977, RMSEA = 0.041 [90% CI = 0.000, 0.086]; and in W6, χ^2^ (5) = 10.585, *p* = 0.060, CFI = 0.982, TLI = 0.964, RMSEA = 0.053 [90% CI = 0.000, 0.098]. The factor loadings for each of these estimated CFA models are shown in Table 2.

### 3.2. Structural Models

We estimated four mediation models to analyze the effect of the perceived impact of COVID-19 on comfort food consumption across times (from W3 to W6) mediated by emotional distress. Each of these mediated models was tested, including a direct effect of the perceived impact of COVID-19, as a predictor measured in W1, on comfort food consumption, the criterion variable measured in W3, W4, W5, and W6; thus, the four models were generated. In all models, the indirect effect was through emotional distress, a mediating variable evaluated in W2 (see Figure 1).

We first evaluated Model 1, which included COVID-19 perceived impact as a predictor of the consumption of comfort food measured in W3. This model presented excellent fit, χ^2^ (41) = 73.869, *p* < 0.001, CFI = 0.980, TLI = 0.973, RMSEA = 0.028 [90% CI = 0.017, 0.038]. In Model 1, the direct effect of COVID-19 on comfort food consumption was not significant. We then evaluated Model 2 which included COVID-19 perceived impact as a predictor of comfort food consumption measured in W4. Like the previous model, this model presented excellent fit, χ^2^ (41) = 79.988, *p* < 0.001, CFI = 0.977, TLI = 0.969, RMSEA = 0.030 [90% CI = 0.020, 0.040]. We also found that the direct effect between perceived impact of COVID-19 and comfort food consumption was not significant. Next, we evaluated Model 3, which incorporated the consumption of comfort food at W5 as the outcome. This model also presented excellent fit, χ^2^ (41) = 59.724, *p* = 0.020, CFI = 0.988, TLI = 0.984, RMSEA = 0.021 [90% CI = 0.007, 0.032], and unlike the two previous models, all effects were significant. Finally, we estimated Model 4, which included the consumption of comfort food in W6 as the outcome. Model 4 showed excellent model fit, χ^2^ (41) = 96.204, *p* < 0.001, CFI = 0.967, TLI = 0.956, RMSEA = 0.036 [90% CI = 0.027, 0.045], and all model parameters were significant. Models using the same variables but changing the waves at which emotional distress was measured were also tested and results can be found in Appendix A analysis (Models 5 to 10).

### 3.3. Direct and Indirect Effects on Consuming Comfort Food

Table 3 shows the direct and indirect effects of the four estimated models. Indirect effects were estimated using bootstrapping procedures and the Monte Carlo test [46]. Models 1 and 2 did not show a direct effect of the perceived impact of COVID-19 (W1) on comfort food consumption in W3 and W4 (*p* > 0.05). Thus, the effect of COVID-19 (W1) on the consumption of comfort food (W3, Adjusted R^2^ = 0.16 and W4, Adjusted R^2^ = 0.12) was mediated entirely through the increase in emotional distress (W2). That is, the initial impact of COVID-19 is indirect through an increase in emotional distress (symptoms of stress, anxiety, and depression), which in turn increased the consumption of comfort food during W3 (Adjusted R^2^ = 0.23) and W4 (Adjusted R^2^ = 0.18). Meanwhile, Models 3 and 4 revealed that the indirect effect of emotional distress on the relationship between the perceived impact of COVID-19 on comfort food consumption declines in W5 and W6, implying that the effect is now partial and not fully mediated. Furthermore, Models 3 and 4 showed that the direct impact of COVID-19 on comfort food consumption is now statistically significant. The direct and indirect impact of alternative models across times (Models 5 to 10) are shown in the Appendix A.

## 4. Discussion

The results obtained in the present study highlight the longitudinal relationships between the perceived impact of the COVID-19 pandemic (economic, interpersonal, and health) and comfort food consumption during a six-month period (from July to December 2020). In particular, we found that the emotional distress generated by the pandemic acted as a long-lasting mechanism that explained the negative perceived impact of COVID-19 on increased comfort food consumption, first completely and then partially.

Specifically, we found that COVID-19 perceived negative impact measured in W1 (July) was indirectly associated with more comfort food consumption in W3 (September) and W4 (October). Thus, our findings suggest that emotional distress fully mediates the relationship between the perceived negative impact of COVID-19 and comfort food consumption during this period. Emotional distress seems to be the primary mechanism explaining the increase of comfort food consumption on W3 and W4. Our findings are consistent with previous research during the H1NI pandemic [35,36,37] that has recognized that threats, such as the pandemic, trigger emotional distress, which results in coping mechanisms. More importantly, our results longitudinally replicate and expand the findings of Ramalho et al. [38] in Portugal. In their cross-sectional study, the authors found that participants who experienced increased psychological distress, due to COVID-19 lockdowns, engaged in more disordered eating behaviors. With this in mind, comfort food consumption can be considered as a coping strategy that has been widely executed during home confinement to deal with the adverse impact and psychological distress produced by the COVID-19 pandemic [16,19,51]. Furthermore, this full mediation effect of emotional distress on the relationship between the perceived impact of COVID-19 and comfort food consumption is strong enough to maintain its influence on this behavior for the following two months.

Unlike our results for W3 and W4, the prediction of comfort food consumption in November and December was jointly explained by the perceived negative impact of COVID-19 in July and its effect on emotional distress in August. This raises the question as to why emotional distress fully mediates the relationship between the perceived negative impact of COVID-19 at W1 and comfort food consumption at W3 and W4 and only partially mediates the same relationship at W5 and W6. One possible explanation for our results is that the emotional distress caused by the pandemic has been regulated and has diminished over time, thus, the associations are not as strong as at the beginning of the study, when there was greater comfort food consumption. This implies that since November (W5), the increase in comfort food consumption would not only be caused by emotional distress but also by the negative perceived impact of COVID-19. This finding is consistent with the development of emotion regulation processes and adaptation processes as a response to the highly stressful situation generated by the pandemic [52,53]; also, with the adoption of coping strategies focused on gratification-orientated behaviors, such as comfort food consumption [23,24]. As previous research has detected, eating comfort food during the pandemic is highly likely due to the restrictive measures implemented to reduce the transmission of the SARS-CoV-2, which include the prohibition of social gatherings and parties, meeting friends, eating in restaurants, and doing outdoor activities and exercise [15,18,21]. Thus, consuming comfort food could be a behavior adopted to manage the negative experience derived from prolonged confinement and cope with the lack of activities and boredom [18,29,30].

Our findings support Lazarus and Folkman’s stress and coping model [34]. We provide longitudinal evidence that recognizing a thread, such as the pandemic, triggers emotional distress covering several adverse emotional reactions (i.e., stress, anxiety, and depression) [1,10,54]. Furthermore, this same emotional reaction generates a behavioral activation or coping mechanism that diverts and sparks positive emotions and feelings of well-being, such as eating comfort food [39,55]. These coping activities act to reduce the psychological burden imposed by emotional distress, releasing cognitive resources to adjust and deal with everyday activities, which have not stopped during the COVID-19 pandemic [51,56]. Our mediational longitudinal model shows that the perceived negative impact of COVID-19 in July, when Chile reported the peak of COVID-19 cases, had a long-lasting effect on comfort food consumption through increased emotional distress. That said, our findings reveal that crises impacting at an economic, interpersonal, and health level (i.e., earthquakes, hurricanes, pandemics) have psychological and behavioral consequences over time [57].

The present study had several strengths. The first is that we used longitudinal data to examine the mediation effect of emotional distress on the relationship between the perceived negative impact of the COVID-19 and comfort food consumption across six waves. Second, we used robust measures and methods to deal with missing values, a common problem when using longitudinal panel data [58]. Third, our sample included participants of different ages, so our model has potentially generalizable results. Fourth, we expanded the cross-sectional findings of Ramalho et al. [38] in Portugal using longitudinal data. Thus, we moved towards the estimation of these over time effects, and also, were able to find similar results on a different continent. And fifth, our model provides guidelines for future interventions to reduce comfort food consumption (e.g., using telehealth), which should focus on handling emotional distress [59].

Despite its strengths, there are some limitations to the current study. First, the study was conducted through online surveys, due to the pandemic; using online surveys might restrict the possibilities of generalizing our findings to people without access to the internet. Second, our sample was limited to students and university staff. Although university staff have maintained their jobs and have been teleworking during the pandemic (and thus, have not been directly exposed to the economic effects of the pandemic), the families of the students may have been affected, and their ability to provide economic support may have changed. Further, as previous research has shown, our sample consisted mainly of women, who are more likely to self-select and participate more than men in surveys (paper-based and online, see Smith, 2008 [60]). This context-specific situation, along with the sample characteristics, limits the use of our findings for broader generalization to more heterogeneous and diverse populations. Third, this study started in July, which corresponds to the winter season in the southern hemisphere. There might be some other contextual factors (such as seasonal food consumption) related to the season, which explain the increase in emotional distress and comfort food (high in calories) consumption (see more in Yang et al. [61]). Fourth, the scale used to measure the perceived negative impact of COVID-19 showed only appropriate construct consistency. Considering that it is a new 3-item scale and that we are in an ongoing emergency, context-specific new items are required to assess the perceived impact of COVID-19. Finally, our longitudinal model could have been more robust if we had measured the perceived negative impact of COVID-19 in all the waves. Thus, we would have been able to control for the autoregressive effects of each variable and to explore cross-lagged mediation [62]. Moreover, future research should not be limited to examining negative emotions as a consequence of the COVID-19 pandemic. As Fredrickson et al. [55] and Ong et al. [63] have stated, threat and stress situations trigger negative and positive emotions together. Thus, future research should consider evaluating positive emotions such as inspiration, enthusiasm, and pride to mediate the relationship between threatening events and coping behaviors.

## 5. Conclusions

To conclude, in this longitudinal research, we were able to reveal the role of emotional distress as a key mechanism to explain coping behaviors, such as comfort food consumption, adopted as a consequence of the economic, interpersonal, and health impact of the COVID-19 pandemic. As previous research has shown, people have adopted unhealthy behaviors to cope with the emotional distress during the pandemic, such as comfort food consumption [16,17]. This type of behavior, along with low physical activity, due to home confinement [19], is a predictor of several health problems, such as eating disorders [64,65], and metabolic diseases [66], including obesity [67], cardiac alterations [68], and diabetes [69]. Assessment of these health outcomes was beyond the scope of our study. However, we would like to alert the university authorities and medical staff to this possibility for consideration in developing preventive strategies focused on mental health.

## Figures and Tables

**Figure 1 nutrients-13-01910-f001:**
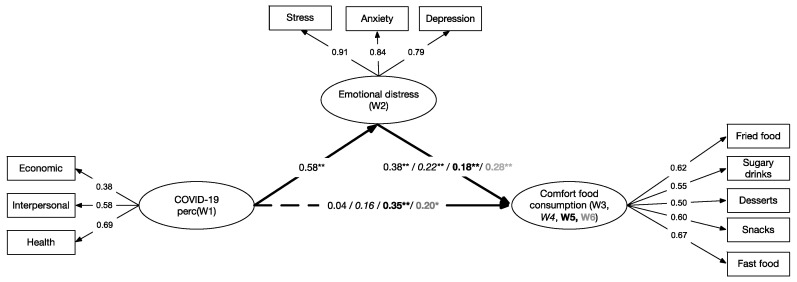
Summary model assessing the standardized regression coefficients of COVID-19 perceived impact in W1 (predictor) through emotional distress in W2 (mediator) on comfort food consumption (outcome) in W3, W4 (italics), W5 (bold) and W6 (grey). Dashed lines represent non-significant associations. Note. * *p* < 0.05, ** *p* < 0.01.

**Table 1 nutrients-13-01910-t001:**
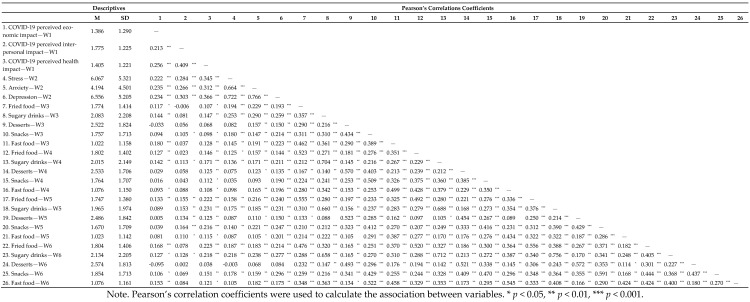
Means (M), standard deviations (SD), and correlations between COVID-19 perceived impact, emotional distress, and comfort food consumption over time.

**Table 2 nutrients-13-01910-t002:** Standardized coefficients and standard errors obtained in the confirmatory factor analysis.

Latent Variable/Indicators	Standardized Coefficient	Standard Error
COVID-19 perceived impact–W1		
Economic	0.436 **	0.048
Interpersonal	0.572 **	0.051
Health	0.706 **	0.060
Emotional distress–W2		
Stress	0.904 **	0.252
Depression	0.812 **	0.239
Anxiety	0.824 **	0.210
Comfort food consumption–W3		
Fried food	0.613 **	0.077
Sugary drinks	0.531 **	0.111
Desserts	0.494 **	0.090
Snacks	0.604 **	0.096
Fast food	0.663 **	0.069
Comfort food consumption–W4		
Fried food	0.554 **	0.074
Sugary drinks	0.506 **	0.109
Desserts	0.449 **	0.085
Snacks	0.682 **	0.105
Fast food	0.628 **	0.069
Comfort food consumption–W5		
Fried food	0.560 **	0.075
Sugary drinks	0.608 **	0.110
Desserts	0.451 **	0.090
Snacks	0.643 **	0.101
Fast food	0.488 **	0.059
Comfort food consumption–W6		
Fried food	0.704 **	0.090
Sugary drinks	0.601 **	0.123
Desserts	0.451 **	0.093
Snacks	0.640 **	0.103
Fast food	0.530 **	0.062

Note. ** *p* < 0.01.

**Table 3 nutrients-13-01910-t003:** Direct, indirect effects, confidence intervals, standard errors, and mediation effects.

Direct Effect		Indirect Effects				
	COVID-19 Perceived impact (W1) → Comfort food (W3, W4, W5, W6)	COVID-19 Perceived impact (W1) → Emotional distress (W2) → Comfort food (W3, W4, W5, W6)	CI 95%	Standard error	Mediation %	Mediation type
Model 1	0.041	0.220	[0.113, 0.332]	0.054	84	Full
Model 2	0.164	0.128	[0.019, 0.233]	0.052	44	Full
Model 3	0.359 **	0.107	[0.002, 0.211]	0.053	22	Partial
Model 4	0.202 *	0.160	[0.052, 0.267]	0.055	44	Partial

Note. Model 1 considers comfort food measured on W3, Model 2 on W4, Model 3 on W5, and Model 4 on W6. * *p* < 0.05, ** *p* < 0.01.

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
