# Peer review of "The Perceived Impact of COVID-19 on Comfort Food Consumption over Time: The Mediational Role of Emotional Distress"

_nutrients, 2021, doi:10.3390/nu13061910_

Round 1
Reviewer 1 Report
This study investigated the strength of associations between the food consumption and the perception of interpersonal, economic, and health implications of COVID 19, and furthermore if these associations were mediated by emotional distress. They show the correlations between the perceived impact, food consumption, and emotional distress (Table 1), the strength of the effect of the indicators on the latent variables and precision (Table 2), a summary model describing the standard regression coefficients of the predictor, mediator, and outcome in the various “waves” (Figure 1) and lastly, these authors summarized the direct and indirect effects evaluated with each model (Table 3). Although the direction of this work is exciting and impactful, several weaknesses need to be addressed:
Weaknesses
- Need to both identify the CR threshold and provide rationale/support for threshold
- Table 1 format is unclear. For example, are the correlation values regression coefficients? Also, what time frames are measured across time?
- The data suggests a diminishing relationship between the perception of COVID 19, emotional distress, and consumption of food. This suggests that a persisting psychological and behavioral effect may not be accurate.
- Discussion of limitations does not address the over-representation of women in this study nor the potential ethnic homogeneity
- Discussion also does not address possible cultural influences on food consumption, particularly in the fall and winter months
- Introduction of applied abbreviations and consistency in terminology is needed.
Author Response
Dear Reviewer,
Please see the attached document.
Thank you.

Reviewer 2 Report
The manuscript is an interesting evaluation of the effects of the COVID19 pandemia on eating food consumption in the general population. I think the authors have described methods and results clearly and the paper is well written.
I have only few comments for the authors:
- Is it possible to have a description of the population through the different waves? The last wave is significantly smaller than the first one, could be this influence the analysis?
- How did you decide these specific questionnaires?
- have you considered results in eating disorder patients? I think that Monteleone et al 2021 (https://doi.org/10.1016/j.jad.2021.02.037) is useful for the scope of the authors to stress the idea that stress events could have a negative impact on eating behaviors.
- Have you any suggestion about clinical impact of you results? I think this could be an interesting conclusion of your paper.
Author Response
Dear reviewer,
Please see the attached document.
Thank you.

Round 2
Reviewer 1 Report
All concerns have been addressed adequately.